

# Intraspecific body size variation and allometry of genitalia in the orb-web spider—*Argiope lobata*

Chathuranga Dharmarathne[1,*], Donald James McLean[1], Marie E. Herberstein[1] and Jutta M. Schneider[2,*]

[1] School of Natural Sciences, Macquarie University, North Ryde, New South Wales, Australia
[2] Department of Biology, Universität Hamburg, Hamburg, Germany
[*] These authors contributed equally to this work.

## ABSTRACT

The current consensus is that sexual selection is responsible for the rapid and diverse evolution of genitalia, with several mutually exclusive mechanisms under debate, including non-antagonistic, antagonistic and stabilizing mechanisms. We used the orb-web spider, *Argiope lobata* (Araneidae), as a study model to quantify the allometric relationship between body size and genitalia, and to test for any impact of genital structures on male mating success or outcome in terms of copulation duration, leg loss or cannibalism. Our data do not support the 'one-size-fits-all' hypothesis that predicts a negative allometric slope between genitalia and body size. Importantly, we measured both male and female genitalia, and there was no sex specific pattern in allometric slopes. Unexpectedly, we found no predictor for reproductive success as indicated by copulation duration, cannibalism, and leg loss.

# INTRODUCTION

Body size has an enormous impact on practically every trait of an animal's morphology, physiology, and ecology (*Lindstedt, 1987*). In its broadest sense, allometry describes how morphological features change according to body size—in other words, it describes the scaling relationship between the size of a trait with the overall size of the body as trait and body growth during development (*Small, 2012*). Arthropods, encompassing diverse groups such as insects, spiders, and crustaceans, exhibit a wide range of allometric patterns of traits such as genitalia and weapons, ranging from positive to negative allometries against body size (*Wickman & Karlsson, 1989*; *Karim, Guild & Thummel, 1993*; *Stern & Emlen, 1999*; *Shingleton et al., 2007*; *Bertin & Fairbairn, 2007*; *Lease & Wolf, 2010*; *Bidau, Taffarel & Castillo, 2016*). In *Argiope aurantia* orb-web spiders, for instance, male and female genital characteristics scale negatively with body size (*Assis & Foellmer, 2016*). The nature of these allometric relationships can provide significant information about trait evolution and function.

Most complex morphologies show allometric variation that can include (i) no relationship with body size (the slope of a log–log regression is not different from 0),

Corresponding author
Chathuranga Dharmarathne,
Chathurangadharma@gmail.com,
chathuranga.dharmarathne@students.
mq.edu.au

(ii) traits which increase in proportion with body size (isometry, slope is not different from 1), (iii) traits which grow disproportionately larger with body size (positive allometry or hyperallometry, slope > 1) or (iv) traits which grow disproportionately smaller with body size (negative allometry or hypoallometry, slope < 1; *Gould, 1966*).

Possibly the most intensely studied morphological traits and their relationship with body size are animal genitalia. So far, genital evolution research has focused primarily on males, with male genitalia receiving around twice the amount of research attention than female genitalia (*Ah-King, Barron & Herberstein, 2014*). To investigate the nature of selection on sexual traits such as genitalia, many studies interpret the allometric relationship between body size and genital size (*see Bonduriansky, 2007*). Explaining genital variation and the selective processes responsible for this variation has been hotly debated with several divergent views (*Eberhard, 1985*; *Edwards, 1993*; *Arnqvist & Danielsson, 1999*; *Kinahan et al., 2008*). While there is current consensus that sexual selection is responsible for the rapid and diverse evolution of genitalia (*Eberhard, 1985*; *Eberhard, 2010*; *Arnqvist, 1998*; *Hosken & Stockley, 2004*, for examples see: *House & Simmons, 2003*; *Simmons & Garcia-Gonzalez, 2011*), there are several mutually exclusive mechanisms under debate. First, male genitalia may be under sexual selection to stimulate the female or to position sperm in locations that facilitate successful fertilisation within the female tract (*Lande, 1981*), thus genitalia evolve in a non-antagonistic process. Second, antagonistic selection resulting from a coevolutionary arms race between male and female reproductive optima may be at play. Males and females often have competing interests when it comes to mating and fertilization, which can lead to the evolution of male genitalia that bypass female choice. This can, in turn, lead to female genitalia counter-evolving defences to regain control over mating or fertilization (at a cost to the male) (*Rice, 1984*; *Arnqvist & Rowe, 1995*; *Arnqvist & Rowe, 2005*; *Rönn, Katvala & Arnqvist, 2007*; *Foerster et al., 2007*; *Kuntner, Coddington & Schneider, 2009*; *Brennan, Clark & Prum, 2010*; *Innocenti & Morrow, 2010*; *Perry & Rowe, 2015*). The third process is a stabilizing mechanism known as 'one-size-fits-all', which argues that males may be selected on intermediate-sized genitalia regardless of body size to be able to copulate with all females regardless of size (*Eberhard et al., 1998*).

*Argiope lobata* is an excellent model for studying allometric relationships of male and female genitalia as they interact both synergistically and antagonistically during copulation. The two copulatory openings of the epigyne are separated by a median septum (Fig. 1) (*Levi, 1968*; *Uhl, Nessler & Schneider, 2007*; *Foelix, 2011*). During copulation, several structures of the male genitalia (the conductor and the pedipalp apophysis) contact the copulatory opening and the median septum (*Schneider, Uhl & Herberstein, 2015*). Moreover, the pedipalp apophysis has a thin and pointed spur, which is common in the *Argiope* genus (*Levi, 1983*). The spur is important for the successful transfer of sperm, and in *A. lobata* removal of the spur prevents successful coupling (*Schneider, Uhl & Herberstein, 2015*).
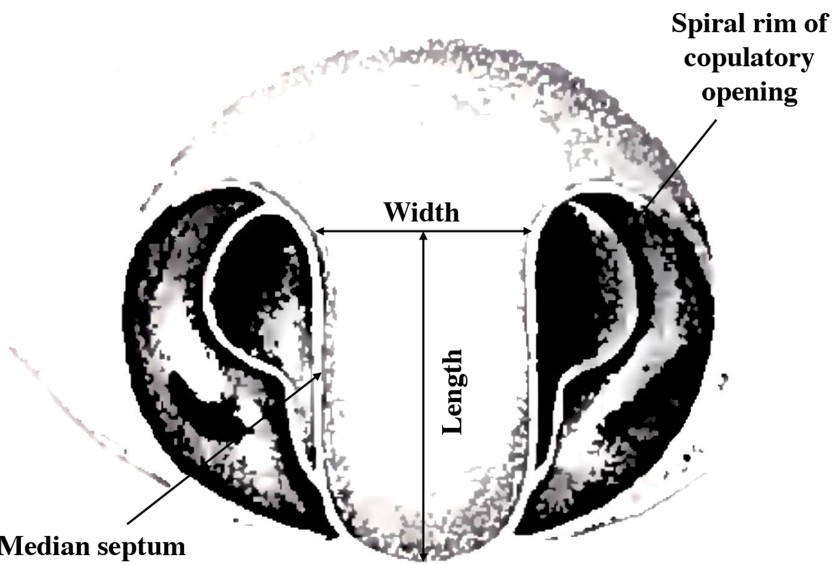

**Figure 1** **Female genital structure characteristics of *A. lobata* (modified diagram from *Levi, 1983*).**

# MATERIAL AND METHODS

Subadult male and female *Argiope lobata* (*Pallas, 1774*) were collected from their natural webs in several sites near Ashkelon, Israel, in 2010. F1 offspring derived from the above collected females were used in this study (Fig. 2). *Argiope lobata* belongs to the family Araneidae and shows sexual dimorphism where females (body length: 16.5–25.5 mm) are larger than males (body length: 5–7.5 mm) (*Preston, 1998*). In the laboratory, spiders were housed in 330 ml plastic containers and were fed with *Drosophila* twice a week. The spiders and their webs were sprayed with water every day. Adult females were placed in Perspex frames ($36 \times 36 \times 6$ cm) after their final moult, where they constructed their customary orb webs. Adult males were housed in plastic containers until the mating experiment started. Here we ask two questions: first, what is the allometric relationship between body size and genitalia and second, do genital structures influence male mating success? We predict that if genital structures are disproportionately larger relative to body size (allometric regression slope is > 1) then genital structures are under positive sexual selection, while if the slope is < 1 they are more likely to be under stabilizing selection ('one size fits all').

Experiments were conducted in a laboratory at the University of Hamburg from June 2010 until September 2010. All spiders used for this study were unmated. Once a male was introduced to a female, the male typically walked slowly through the female's web towards the hub. As characterised in *A. keyserlingi* (*Wignall & Herberstein, 2013*; *Wignall & Herberstein, 2022*), fast vibrating movements were commonly observed during the courtship. The female's receptivity was frequently indicated by a slow tugging of the web and a mating position in the hub where her abdomen was slanted backwards, providing access to the copulatory openings. The male climbed onto the female, drumming her abdomen with his pedipalps until inserting one of his pedipalps into her copulatory

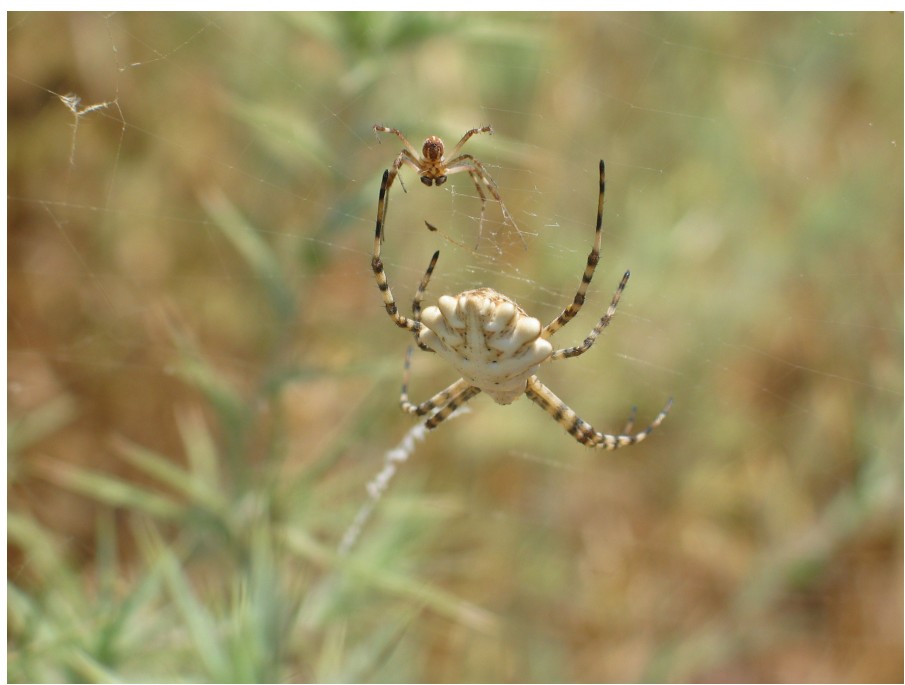

**Figure 2** *Argiope lobata* **male and female in their natural habitats at Ashkelon, Israel.**

opening. As copulation commenced, the female typically attacked the male thereby ending the copulation and her attack frequently resulted in either cannibalism or leg loss (*Nessler, Uhl & Schneider, 2009*). Copulation duration (from the point of pedipalp insertion to the removal of the pedipalp from the copulatory opening of the female) (seconds), frequency of female attacks, cannibalism and leg loss during copulation were recorded. If the male was still alive after copulation, he was classified as 'not cannibalized'. Cannibalized males were removed from the female and saved for further investigations.

After copulation, males and females were euthanized, and body measurements were taken. Leg length and the carapace maximum width and length were measured (to the nearest 0.01 mm) as indicators of body size (*see* also *Nessler, Uhl & Schneider, 2009*). We then calculated carapace area from the width and length and square root transformed area for analysis. Most studies that investigate sperm transfer in spiders used either body mass, tibia and patella length (*Dharmarathne & Herberstein, 2022*) or the square root of the carapace area as proxy for body size (*Assis & Foellmer, 2016*). Because body mass can vary substantially between individuals of similar size due to their recent feeding status, we measured both the carapace area (later converted to square root carapace area) and total length of the tibia and patella of the first pair of legs as a proxy of body size, using a stereomicroscope (Fig. 3).

We measured several male and female genital structures that are known to interact during copulation or that are linked to sperm transfer. We measured female median septum length and width under a stereomicroscope. Length and width were measured at the longest and widest points of the median septum respectively. In male pedipalps,

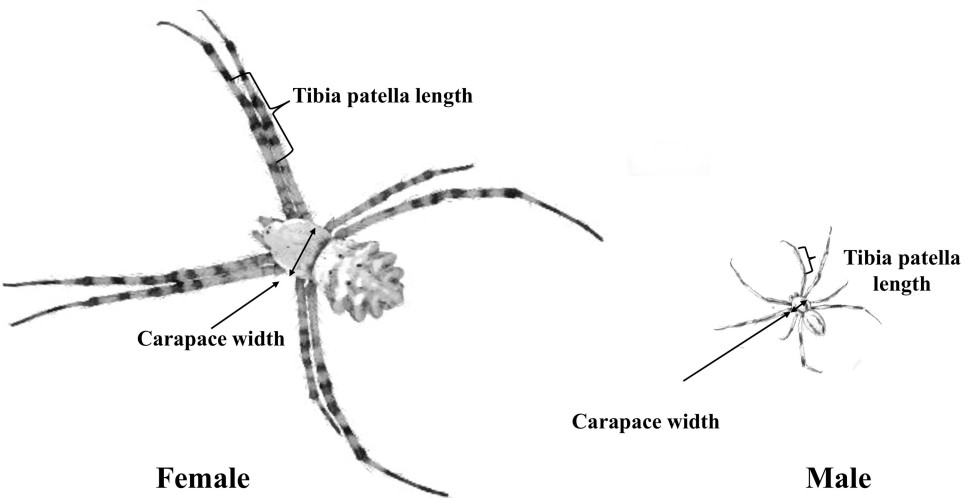

**Figure 3** **Male and female body structure characteristics of *A. lobata*.**

the apophysis connects tightly with the female genital structures, especially with the median septum, during sperm transfer (*Huber, 1995*; *Welke & Schneider, 2009*; *Hirt, Ruch & Schneider, 2017*). The presence of a slender, pointed spur on the apophysis is unique to the *Argiope* genus and the size and shape of the spur varies between species (*Levi, 1983*). The spur aids in the process of genital damage where the tip of the embolus breaks off during copulation (*Nessler, Uhl & Schneider, 2007*). In this study, we measured the area of the apophysis and the length of the spur under a stereomicroscope. Microscopic analysis was conducted using a Leica MZ16 stereomicroscope (Leica Microsystems, Wetzlar, Germany) equipped with a Leica DFC320 digital camera (Leica Microsystems, Wetzlar, Germany). The microscope was connected to a computer running Leica IM500 image analysis software (Leica Microsystems, Wetzlar, Germany) for capturing and processing the acquired images. Length was measured in µm for all measurements. We discarded records involving one female with median septum width $< \frac{1}{2}$ that of the other spiders (ID Al1 570), and one male with an abnormally short right tibia-patella length (ID Al6 653).

## Allometric relationships

Testing allometric hypotheses usually involves fitting a linear model to log-transformed measures of body size and the trait of interest (*Warton et al., 2006*). Choice of line-fitting method is the subject of an extended and robust debate (*e.g.*, *Eberhard, Huber & Rodriguez, 1999*; *Eberhard et al., 1998*; *Green, 1999*; *Kilmer & Rodríguez, 2017*; *Smith, 2009*; *Warton et al., 2006*), however there is no single correct choice, as the appropriate method depends on the biological question (*Smith, 2009*; *Warton et al., 2006*). Ordinary least squares regression (OLS) identifies the variation in a variable *y* that can be described as a function of another variable *x*. Accordingly, it is appropriate for assessing variation in a trait *y* (such as genital size) that is caused by variation in a trait *x* (such as body size), while allowing for "error" in *y* that arises from other factors (*Kilmer & Rodríguez, 2017*). Here, the term "error"

encompasses all causes of variation in $y$ that are not accounted for (or caused) by the variation in $x$; importantly this "error" includes biological and environmental factors ("equation error"; *Warton et al., 2006*), as well as experimental and measurement error ("measurement error"; *Warton et al., 2006*) (arguably, "noise" may be a more appropriate and less misleading term than "error"). If, however, trait $x$ and trait $y$ are both explained by a third trait, $z$, and the errors in $x$ and $y$ are independent, then OLS will underestimate the slope of the relationship, and the greater the error in $x$, the greater the underestimate (*Smith, 2009*). *Smith (2009)* describes this situation as a "symmetric relationship" and finds that reduced major axis (RMA) regression is indicated here. RMA is also known as "standardised major axis" (SMA). In our tests of allometry, we use proxy measures of body size, the square root of the carapace area and tibia-patella length. Hence, we use RMA for estimating the slope of the regression line when testing allometric hypotheses.

When testing for hyperallometry or isometry, we are asking how one variable (*e.g.*, genital size) increases with another (such as body size), because we are looking for a functional scaling relationship between the two variables. For reasons described above, we estimated the regression slope using RMA, however RMA does not take covariance between the two variables into account. Hence, when testing for isometry or hyperallometry, we must test two conditions: (1) the RMA slope is equal to (isometry) or significantly greater than 1 (hyperallometry), and (2) there is a positive correlation between the two variables. To test condition 2, we test whether the OLS regression slope is significantly greater than 0 (*Warton et al., 2006*). When testing the one-size-fits-all hypothesis, we need to determine whether there is less variation in genital size than would be expected from simple isometry. In this case, the nature of the relationship between body size and genital size (*i.e.,* the existence or otherwise of a causal relationship, or even whether there is a correlation) is immaterial. For that reason, OLS is not suitable and again, RMA regression is the appropriate choice (*Smith, 2009*; *Warton et al., 2006*). For hypothesis testing, we compared the 99% confidence intervals of the regression slopes to 1 (rather than the more conventional level of 95%) since we performed multiple comparisons (*Altman et al., 2000*). Accordingly, we tested for statistical significance with the corresponding level of alpha = 0.01. Analyses were performed in R (*R Core Team, 2020*) using the packages smatr (*Warton et al., 2012*), boot (*Canty & Ripley, 2002*) and Durga (*Khan & McLean, 2023*).

## INFLUENCE OF BODY SIZE AND GENITAL MORPHOLOGY ON MALE REPRODUCTIVE OUTCOME

We only included data where individuals copulated. Impact of morphology on copulation duration was determined by fitting a multiple linear regression with body size and genital characteristics as independent variables and copulation duration as the dependent variable. Similarly, the impact of morphology on leg loss or cannibalism during copulation was assessed by fitting a generalized linear model to the data with a logit error distribution.

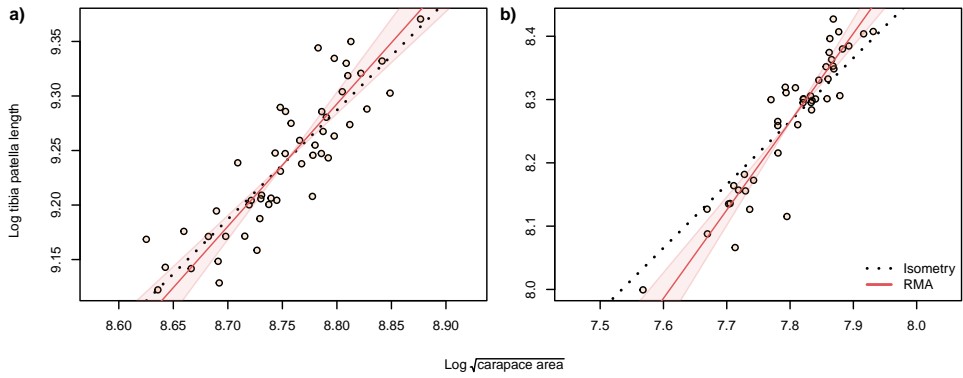

**Figure 4** **Static allometry of $\sqrt{carapace\ area}$ and tibia patella length, in female (A) and male (B) *A. lobata*.** Red line shows the fitted RMA regression. Filled polygon depicts 99% confidence intervals (CI) of the RMA slope. The black dotted line indicates isometry (slope = 1 and passing through the centroid of the points). Points represent individual spiders. Female leg length is isometric with carapace area, male leg length is hyperallometric.

## RESULTS

When analysing allometric relationships, both the square root of carapace area and tibia-patella length have been used as proxies for spider body size (see *Dharmarathne & Herberstein, 2022* for a review). The square root of carapace area may be a more relevant linear indicator of body size (*Assis & Foellmer, 2016*) for sexually dimorphic species such as *Argiope lobata,* where leg length in males may also be under selection. Consequently, we first analysed the scaling relationship of patella-tibia length with the square root of carapace area and found that male tibia-patella length exhibited hyperallometry with respect to the square root of carapace area, while female tibia-patella length was isometric (Fig. 4). Accordingly, we performed the allometric relationship analyses between genital traits and body size using both the square root of carapace area and tibia-patella length as body size proxies.

### Allometric relationships

The slope of the log–log RMA regression against tibia-patella length and square root carapace area was used to determine whether the genital structures are under directional selection (hyperallometry, slope > 1), stabilizing selection ('one size fits all': hypoallometry, slope < 1) or scale in proportion to body size (isometry, slope = 1). To establish isometry or hyperallometry, we also required a significant positive correlation between the variables. Of the four traits we compared, only a single RMA slope differed significantly from 1: female median septum width, with a slope >1 (Tables 1 and 2). Of the four traits, only pedipalp apophysis area showed a significant correlation with body size (tibia patella length and the square root of the carapace area), so no genital traits satisfied our criteria for either hyperallometry or hypoallometry (Figs. 5 and 6).

**Table 1** Static allometry: 99% confidence interval (lower limit, LL, and upper limit, UL) of slope of reduced major axis (RMA) regression analysis and correlation (*P*-value and R²) of male and female genital characteristics on body size proxy.

| Body size *vs* genital characteristics | LL | UL | *P* | R² | *n* |
|---|---|---|---|---|---|
| Female TPL *vs* median septum length | 0.88 | 1.80 | 0.02 | 0.11 | 50 |
| Female TPL *vs* median septum width | 1.06 | 2.24 | 0.21 | 0.03 | 50 |
| Male TPL *vs* pedipalp apophysis area | 0.62 | 1.27 | 0.002 | 0.21 | 47 |
| Male TPL *vs* spur length | 0.57 | 1.24 | 0.25 | 0.03 | 47 |

**Table 2** Static allometry: 99% confidence interval (lower limit, LL, and upper limit, UL) of slope of reduced major axis (RMA) regression analysis and correlation (*P*-value and R²) of male and female genital characteristics on body size proxy ($\sqrt{carapace\ area}$, SCA) in *A. lobata*. LL and UL values in bold indicate slope $\neq$ 1. *P*-value and R² in bold indicate a significant positive correlation between SCA and genital trait at the $p < 0.01$ level.

| Body size *vs* genital characteristics | LL | UL | *P* | R² | *n* |
|---|---|---|---|---|---|
| Median septum length *vs* female SCA | 0.99 | 2.02 | 0.013 | 0.12 | 50 |
| Median septum width *vs* female SCA | 1.19 | 2.51 | 0.023 | 0.03 | 50 |
| Pedipalp apophysis area *vs* male SCA | 0.89 | 1.73 | <0.001 | 0.29 | 47 |
| Spur length *vs* male SCA | 0.89 | 1.93 | 0.066 | 0.07 | 47 |

## Influence of body size and genital morphology on male reproductive outcome

Multiple linear regression analysis was conducted to determine the impact of body size and genital characteristics on copulation duration. No variables were found to significantly affect copulation duration (Table 3). Further, logistic regression models were used to determine the impact of size and genital morphology on the copulation outcomes, cannibalism or leg loss. Again, none of the size and morphology traits had a significant impact on cannibalism or leg loss (Tables 4 and 5).

Testing our four allometric hypotheses involve eight analyses of statistical significance, leading to a high family-wise rate of type I errors if not controlled for. Applying a Bonferroni correction for eight comparisons results in an alpha level of $0.05/8 = 0.00625$, which is very conservative (*Nakagawa, 2004*). We instead selected the more statistically powerful alpha level of 0.01, as it is easy to comprehend and visualize the corresponding 99% confidence intervals (*i.e.,* 99% CI corresponds to alpha = 0.01), and is recommended by *Altman et al. (2000)*. We feel that applying an alpha of 0.05 would result in an unacceptably high probability of type I errors.

## DISCUSSION

There are several ways of performing allometric studies within a species, including plotting allometric relationships between size and genital traits (*Arnqvist, 1997*) or conducting an experiment that links genital traits with direct selection outcomes. Here, we apply both approaches by investigating the nature of the allometric relationship between body size and genitalia and asking whether these genital structures influenced male mating success in

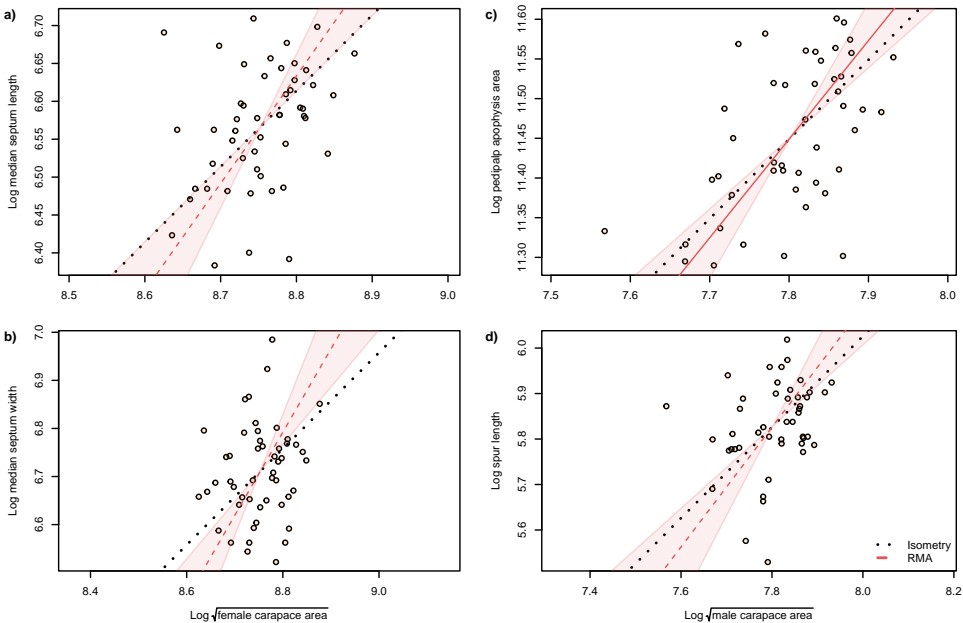

**Figure 5** Static allometry of genital structures and body size proxy ($\sqrt{carapace\ area}$) in female (left panels) and male (right panels) *A. lobata*. Red line shows the fitted RMA regression; dashed indicates there is not a significant correlation between the two variables. Filled polygon depicts 99% confidence intervals (CI) of the RMA slope. The black dotted line indicates isometry (slope $= 1$ and passing through the centroid of the points). Points represent individual spiders. Evidence for allometry requires (1) a significant correlation between the two variables and (2) the RMA CI does not include the line of isometry. None of the relationships satisfy both conditions.

**Table 3** Regression coefficients for copulation duration *vs* male and female genital size and tibia-patella length in *A. lobata* ($R^2 = 0.04$, $n = 32$, $p = 0.98$).

| Variable | Estimate | Std. error | *t* value | Pr (> |*t*|) |
|---|---|---|---|---|
| (Intercept) | $-351.2$ | 400.3 | $-0.87$ | 0.39 |
| Median septum length | 0.025 | 0.25 | 0.10 | 0.92 |
| Median septum width | 0.079 | 0.19 | 0.42 | 0.98 |
| Pedipalp apophysis area | 0.001 | 0.002 | 0.52 | 0.61 |
| Spur length | 0.049 | 0.47 | 0.11 | 0.92 |
| Male tibia-patella length | 0.003 | 0.035 | 0.08 | 0.93 |
| Female tibia-patella length | 0.018 | 0.026 | 0.72 | 0.48 |

terms of copulation duration, leg loss and cannibalism. We found no strong evidence for a hyper- or hypoallometric relationship between male and female body size and genital size. These results do not support the "one size fits all" hypothesis, which states that genitalia are under stabilising selection (*Uhl & Vollrath, 2000*; *Ramos et al., 2005*; *Hosken, Minder & Ward, 2005*). Furthermore, the size of genitalia did not affect copulation success. Neither copulation duration, cannibalism nor leg loss were related to body size or the size of genital traits.

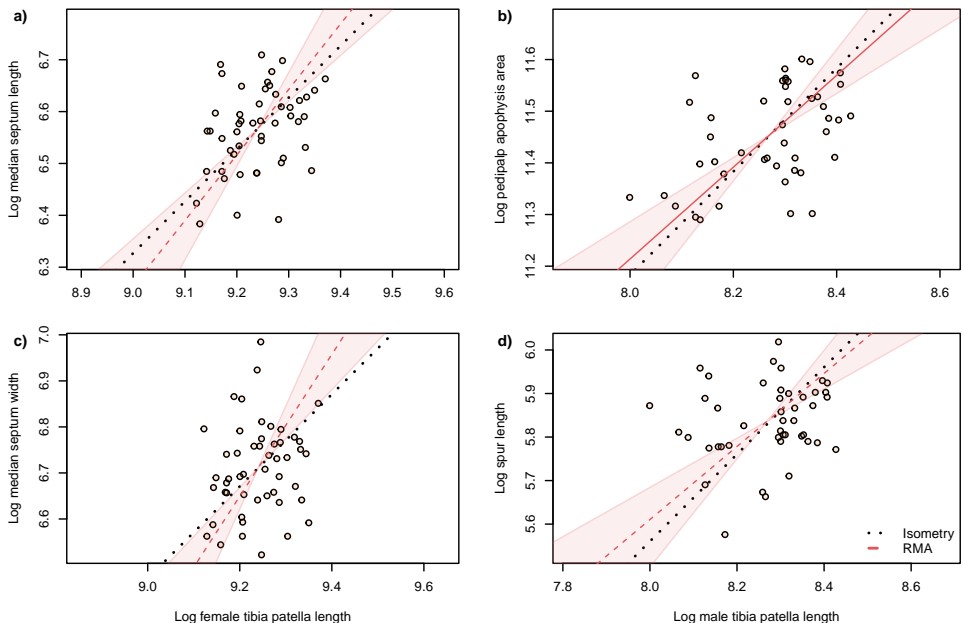

**Figure 6** **Static allometry of genital structures and body size proxy (tibia-patella length) in female (left panels) and male (right panels) _A. lobata_.** Red line shows the fitted RMA regression; dashed indicates there is not a significant correlation between the two variables. Filled polygon depicts 99% confidence intervals (CI) of the RMA slope. The black dotted line indicates isometry (slope = 1 and passing through the centroid of the points). Points represent individual spiders. Evidence for allometry requires (1) a significant correlation between the two variables and (2) the RMA CI does not include the line of isometry.

**Table 4** **Logistic regression coefficients for cannibalism _vs_ male and female genital size and tibia patella length in _A. lobata_. Dispersion parameter for binomial family taken to be 1.**

| Variable | Estimate | Std. error | $z$ value | Pr (> |z|) |
|---|---|---|---|---|
| (Intercept) | 15.6 | 12.9 | 1.2 | 0.23 |
| Median septum length | 0.003 | 0.008 | 0.36 | 0.72 |
| Median septum width | −0.009 | 0.006 | −1.4 | 0.17 |
| Pedipalp apophysis area | −0.00005 | 0.00006 | −0.83 | 0.40 |
| Spur length | −0.002 | 0.012 | −0.14 | 0.89 |
| Male tibia-patella length | −0.0006 | 0.001 | −0.59 | 0.55 |
| Female tibia-patella length | −0.0002 | 0.0008 | −0.23 | 0.82 |

A study on the congener _A. aurantia_ that used a similar approach, also found little support for the "one-size-fits-all" hypothesis and no support for the "lock-and-key" hypotheses. In addition to investigating the nature of the allometric relationship between body size and the size of genitalic traits, they also experimentally tested whether these genital structures influence aspects of male mating success, including sperm transfer success and found no significant correlations (_Assis & Foellmer, 2016_).

Hypoallometric scaling (slope < 1) and rapid diversification of genitalia among species have, in fact, been an evolutionary puzzle. Studies indicating hypoallometric scaling

**Table 5  Logistic regression coefficients for leg loss *vs* male and female genital size and tibia patella length in *A. lobata*. Dispersion parameter for binomial family taken to be 1.**

| Variable | Estimate | Std. error | *z* value | Pr (> |*z*|) |
|---|---|---|---|---|
| (Intercept) | 17.1 | 13.6 | 1.2 | 0.21 |
| Median septum length | 0.0007 | 0.007 | 0.09 | 0.93 |
| Median septum width | −0.004 | 0.006 | −0.63 | 0.53 |
| Pedipalp apophysis area | −0.00007 | 0.00006 | −1.1 | 0.27 |
| Spur length | −0.02 | 0.02 | −0.99 | 0.32 |
| Male tibia-patella length | −0.0003 | 0.001 | −0.32 | 0.75 |
| Female tibia-patella length | −0.0001 | 0.0008 | −0.15 | 0.88 |

(slope < 1) of animal genitalia with body size were described by *Eberhard (2009)*. The more common pattern describes genitalia that are hyperallometric (slope > 1) and thought to be under strong selection (*Lloyd, 1979*; *Eberhard, 1985*; *Eberhard, 1996*; *Eberhard, 2010*; *Hosken & Stockley, 2004*; *Simmons, 2014*, but see *Bonduriansky, 2007*). By contrast, our study found slopes around 1 for *Argiope lobata*. Even the steepest slope for female median septum against tibia-patella length was only slightly greater than 1 and the correlation was not significant. Isometry between genital traits and body size may be more common than expected and could be the result of unidentified trade-offs (*Bonduriansky, 2007*).

*Eberhard et al. (1998)* suggested that in spiders specifically, the 'one size fits all' mechanism (slopes < 1) may be pervasive as it suits the overall sexual size dimorphism and sometimes dimorphic males in spiders. Small males are relatively prevalent among spiders, especially among orb-web spiders (*Hormiga, Scharff & Coddington, 2000*). Moreover, in many spider species, such as *Trichonephila clavipes*, *T. plumipes* or *T. edulis*, males can be highly polymorphic (*Cohn, 1990*; *Elgar & Fahey, 1996*; *Schneider et al., 2000*). For example, in *T. edulis*, large males can be 10 times the body length of small males (*Schneider et al., 2000*). According to the 'one-size-fits-all' concept, males with average sized genitalia, independent of their own body size, should be able to copulate with all females they encounter irrespective of their relative size, particularly when there is substantial variation in male size (*Eberhard, 1985*). While not described as intrasexually polymorphic, *Argiope lobata* nevertheless displays variation in male than female body size (*see* Table A1, *male TPL* CV = 10%; *Zimmer, 2014*). Even though sexual dimorphism in *A. lobata* is pronounced (*Foelix, 2011*, *size measurements in this study*) this species does not seem to fit the expectations of the 'one size fits all' mechanism (see supplementary summary statistics for traits).

Copulation duration as a proxy for reproductive success in spiders is frequently used in studies, where copulation duration is positively associated with the number of sperm transferred in various species (*Schneider et al., 2006*; *Herberstein et al., 2011*; *Albo, Bilde & Uhl, 2013*; *Ceballos, Jones & Elgar, 2015*). While it is reasonable to assume that a greater amount of sperm transferred to the female will also result in more fertilizations for the male, the direct link between the number of sperm transferred and fertilization is difficult to establish, as quantifying sperm in the spermatheca prior to fertilization requires the destruction of the female. One way of overcoming this limitation is to count the number of sperm that remain in the used male pedipalp relative to the number of sperm in the

unused pedipalp as an estimate of sperm transfer (*Schneider et al., 2006*). Alternatively, direct fertilization success using sterile techniques can also link relative copulation duration of two males to fertilization (see *Magris, Wignall & Herberstein, 2020*; *Elgar, Schneider & Herberstein, 2000*). In our study, we found that larger females copulated for longer than smaller females. These longer copulations could be driven by the female, the male or both, reflecting female requirement for more sperm or male investment into more fecund females.

## CONCLUSION

Our data do not support the "one-size-fits-all" hypothesis for genitalia in *Argiope lobata*. Importantly, we measured both male and female genitalia, and there was no sex specific pattern in allometric slopes. Unexpectedly, we found no predictor for reproductive success as indicated by copulation duration, cannibalism, and leg loss. It may be that actual fertilisation success is a more appropriate measure of reproductive success in *A. lobata*, even though copulation duration in *A. keyserlingi* was positively related to fertilisation success (*Magris, Wignall & Herberstein, 2020*). Furthermore, we did not consider internal genital morphology as a contributing factor, which may also be under strong sexual selection, but is rarely studied.

## ACKNOWLEDGEMENTS

We thank Prabath Samarasooriya for help with schematic diagrams. We acknowledge the Wallumattagal clan of the Dharug nation as the traditional custodians of the Macquarie University land. We strongly support equity, diversity and inclusion in science (*Rößler, Lotters & Fonte, 2020*). The authors come from different countries(Sri Lanka, Austria, Australia, Germany) and represent different career stages (PhD student, ECR and Professors). Two of the authors self-identify as a member of the LGBTQI+community. The authors have no conflicts of interest to declare.

## APPENDIX

**Table A1** Summary statistics of spider traits used in the analysis. Linear units are $\mu$m and area units are $\mu$m$^2$.

| Trait | MEAN | SD | CV | *n* |
|---|---|---|---|---|
| Female TPL | 10319.7 | 660.6 | 0.064 | 50 |
| $\sqrt{\text{Female carapace area}}$ | 6338.9 | 358.1 | 0.056 | 50 |
| Median septum length | 713.3 | 56.7 | 0.08 | 50 |
| Median septum width | 825.0 | 82.8 | 0.10 | 50 |
| Male TPL | 3923.2 | 400.9 | 0.10 | 45 |
| $\sqrt{\text{Male carapace area}}$ | 2456.1 | 177.2 | 0.072 | 47 |
| Pedipalp apophysis area | 94529.9 | 8725.2 | 0.092 | 46 |
| Spur length | 341.9 | 32.5 | 0.095 | 46 |

### Funding

The authors received no funding for this work.

### Competing Interests

The authors declare there are no competing interests.

### Author Contributions

- Chathuranga Dharmarathne conceived and designed the experiments, analyzed the data, prepared figures and/or tables, authored or reviewed drafts of the article, data analysis, and approved the final draft.
- Donald James McLean conceived and designed the experiments, analyzed the data, prepared figures and/or tables, authored or reviewed drafts of the article, data analysis, and approved the final draft.
- Marie E. Herberstein conceived and designed the experiments, prepared figures and/or tables, authored or reviewed drafts of the article, and approved the final draft.
- Jutta M. Schneider conceived and designed the experiments, performed the experiments, prepared figures and/or tables, authored or reviewed drafts of the article, and approved the final draft.

### Data Availability

The statistical analysis of the data is available in the Supplemental Files.

### Supplemental Information

Supplemental information for this article can be found online at http://dx.doi.org/10.7717/peerj.16413#supplemental-information.

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
