# Peer review of "Intraspecific body size variation and allometry of genitalia in the orb-web spider—Argiope lobata"

_PeerJ, doi:10.7717/peerj.16413_

## Round 0.1 · original submission · Major Revisions

Dear Dr. Dharmarathne and colleagues:

Thanks for submitting your manuscript to PeerJ. I have now received two independent reviews of your work, and as you will see, the reviewers raised some concerns about the research. Despite this, these reviewers are optimistic about your work and the potential impact it will have on research studying spider morphology and evolution. Thus, I encourage you to revise your manuscript, accordingly, taking into account all of the concerns raised by both reviewers.

Please consider the helpful recommendations by the reviewers for improving the analyses, as well as including more information for assessing morphological traits. Please also clarify the discussion to remove confusion regarding the interpretation of intrasexual dimorphism.

While the concerns of the reviewers are relatively minor, this is a major revision to ensure that the original reviewers have a chance to evaluate your responses to their concerns.

Good luck with your revision,

-joe

Reviewer 1 ·

Basic reporting

No Comment

Experimental design

Methods:
Line 97-98, please include the unit of time used to measure copulation duration. From looking at the raw data, I think it is milliseconds?

Validity of the findings

Results:
Did the authors make any measurements or observations about parings that were not successful? Specfically, I am wondering about especially large females paired with especially small males, or vice versa, given some of the points made in the conclusion (one-size-fits-all hypothesis is that sexual dimorphism can be overcome by maintaining a constant size in genitalia). This may be an interesting data point, but the study is fine without. I suggest making it clear that the study only investigated pairings where copulation occurred.

Conclusions:
Did the authors evaluate their results with the more common alpha of .05? It may be interesting to discuss whether the stringency of their level of significance is the reason for the results they obtained.

Lines 211-219: The discussion of males that are intrasexually dimorphic is confusing as an example when the intrasexual dimorphisim of A. lobata males isn't presented as extreme. Perhaps more discussion of the size differences among A. lobata males would better explain this discussion.
The point that male/female dimorphisim in A. lobata is extreme is clear.

Additional comments

The authors present an interesting study of allometry in genitalia among spiders. It provides a much needed data-point to better understanding the complex influences of structure/function constraints, sexual selection, and antagonistic, non-antagonistic, and stabilizing selection mechanisms of evolution. The manuscript is clear and well-written. It was really fun to read!

·

Basic reporting

The manuscript is well written and the topic (intra-specific scaling of animal genitalia) is adequately introduced, although I think it would be nice to give a few concrete examples of scaling patterns which have been documented in arthropods or state any insight of reviews of scaling relationships.

Experimental design

The mating experiments conducted are straight forward and sound. I suggest adding to the methods the sample sizes, as well as more details regarding the measurements (was an ocular micrometer used?; brand and model of the stereo microscope; some indication of measurement error or repeatability of measurements). The latter is important because of the reported lack of significant correlations between genital traits and patella-tibia length. It would be good to show that the relatively small genital traits could be measured with sufficient precision. I think it would also be informative to give summary statistics for the traits, including the coefficient of variation for the traits.

Validity of the findings

I see a fundamental problem with the analysis; however, the good news is that it can easily be addressed: the authors used patella-tibia length as a proxi for body size. This is as such totally acceptable and there are plenty of published studies which do the same. However, when it comes to analyzing scaling relationships, this can be tricky. It is true that the question of what constitutes the “best” body size metric can almost be philosophical: is it body mass, volume, length, width, some composite (e.g. principal component of several linear traits)? It depends on the organism, the feasibility of measuring, and the question one wants to answer. For spiders, the best easy-to-measure linear trait as a useful indicator of body size is probably carapace width (or length, or, even better, the square-root of the estimated carapace area based of width and length), especially when patterns for males and females are compared. This is because relative leg length is sexually dimorphic in adult spiders, with males having relative longer legs than females, which is probably favored during the period of mate search as males are typically the actively search sex. In addition, it has been shown in at least a few Argiope species that leg length (i.e. patella-tibia length) scales hyper-allometrically with carapace dimensions in males, but isometrically in females (e.g. Assis & Foellmer 2016, Eberhard et al 1998 – you already cite both papers). This makes the interpretation of scaling relationships of genital traits with leg length difficult. BUT, the data set does contain carapace width and length measurements, so I encourage the authors to reassess the scaling relationships using the square-root of carapace area as the body size indicator variable, including the scaling of patella-tibia length with body size (as in e.g. Assis & Foellmer 2016).

Additional comments

I hope you find my comments helpful! Matthias Foellmer, NYC, 24 May 2023

---

## Round 0.2 · Minor Revisions

Dear Dr. Dharmarathne and colleagues:

Thanks for revising your manuscript. One reviewer is very satisfied with your revision; however, reviewer 2 raised some concerns and some edits to make. Please address these ASAP so we may continue consideration of your work for publication.

Thanks,

-joe

Reviewer 1 ·

Basic reporting

No comment

Experimental design

No comment

Validity of the findings

No comment

Additional comments

The authors addressed all of my concerns from the initial review.

·

Basic reporting

Unfortunately, I have to say that this revision of the manuscript is a bit disappointing. Relevant information from previous studies is still ignored and the analysis can still be improved (see comments below).

L241/242: “While not described as intrasexually polymorphic, Argiope lobata nevertheless displays a notable variation in male size than females” -> a greater variation in male size than in female size?

Experimental design

Thanks for providing the requested info - no further comment.

Validity of the findings

The authors state that “we used tibia-patella length as our indicator of spider body size, similar to previous studies”. This implies that studies on *allometry* used patella-tibia length as the general indicator of body size before, which is not the case. This statement should be reworded. In my first review, I pointed out that “it has been shown in at least a few Argiope species that leg length (i.e. patella-tibia length) scales hyper-allometrically with carapace dimensions in males, but isometrically in females”. Following my suggestion, the authors did an additional analysis using carapace area, and this analysis was placed in the supplementary materials because results seem similar. However, they did not analyze the scaling relationship of patella-tibia length with carapace area, as in previous studies. I think this leaves out interesting information, precisely because leg length appears to be important for fitness in males, with males having relatively longer legs than females. Of course, it could be argued that the focus of the present study is on genitalic traits and that leg length does not matter in this context, but the comparison of genitalic traits with somatic ones can only add to the whole picture. Thus, I recommend presenting the analysis using carapace area as the body size estimator in the main body of the text, including the scaling analysis for leg length for comparison.

Discussion, first paragraph: As I mentioned in my previous review, Assis & Foellmer (2016) also employed both approaches, investigating the nature of the allometric relationship between body size and the size of genitalic traits and experimentally testing whether these genital structures influence aspects of male mating success, including sperm transfer success. They found little support for the “one-size-fits-all” hypothesis (and none for the “lock-and -key” hypotheses, which is not considered in this ms). Why these findings for a congener are still completely ignored I do not understand. Please include this in the discussion (and perhaps also intro).

L226-31: I think this statement is still a bit misleading and the authors are (again) leaving out a relevant comparison with the congener, Argiope aurantia:
- There are many studies actually showing hypoallometric scaling (slope < 1) of animal genitalia with body size (see e.g. review in Eberhard 2009 Evolution). Indeed, the combination of hypoallometric scaling and rapid divergence of genitalic traits across species has been an evolutionary riddle. The authors do make now a general statement in the intro: "Arthropods, encompassing diverse groups such as insects, spiders, and crustaceans, exhibit a wide range of allometric patterns of traits such as genitalia and weapons, ranging from positive to negative allometries against body size." But I think it is warranted to highlight the wide-spread occurrence of hypoallometric scaling.
- As an example, in Argiope aurantia, male and female genitalic traits scale hypoallometrically with body size (Assis & Foellmer 2016). I think it is important to contrast this with your finding.

Additional comments

I hope you find these comments helpful!
Matthias Foellmer, Berlin, 22.08.2023

---

## Round 0.3 · accepted · Accept

Dear Dr. Dharmarathne and colleagues:

Thanks for revising your manuscript based on the concerns raised by the reviewer. I now believe that your manuscript is suitable for publication. Congratulations! I look forward to seeing this work in print, and I anticipate it being an important resource for groups studying spider morphology and evolution. Thanks again for choosing PeerJ to publish such important work.

Best,

-joe